

# Genome-wide identification of *DUF506* gene family in *Oryza sativa* and expression profiling under abiotic stresses

Wei Dong[1,*], Jian Tu[1,*], Wei Deng[1], Jianhua Zhang[1], Yuran Xu[1], Anyu Gu[1], Hua An[1], Kui Fan[2], Rui Wang[2], Jianping Zhang[2], Limei Kui[1] and Xiaolin Li[1]

[1] Yunnan Academy of Agricultural Sciences, Food Crops Research Institute, Kunming, China
[2] Yunnan Grain Industry Group Co., Ltd, Kunming, China
* These authors contributed equally to this work.

Corresponding authors
Limei Kui, klm@yaas.org.cn
Xiaolin Li, xiaolinli@163.com

## ABSTRACT

The domain of unknown function 560 (DUF560), also known as the PDDEXK_6 family, is a ubiquitous plant protein that has been confirmed to play critical roles in Arabidopsis root development as well as ABA and abiotic responses. However, genome-wide identification and expression pattern analysis in rice (*Oryza sativa*) still need to be improved. Based on the phylogenetic relationship, 10 *OsDUF506* genes were identified and classified into four subfamilies. Segmental duplication was essential to the expansion of *OsDUF506s*, which were subjected to purifying selective pressure. Except for *OsDUF50609* and *OsDUF50610*, the *OsDUF506s* shared colinear gene pairs with five monocot species, showing that they were conserved in evolution. Furthermore, the conserved domains, gene structures, SNPs distribution, and targeting miRNAs were systematically investigated. Massive cis-regulatory elements were discovered in promoter regions, implying that *OsDUF506s* may be important in hormone regulation and abiotic stress response. Therefore, we analyzed plant hormone-induced transcriptome data and performed qRT-PCR on eight *OsDUF506s* under drought, cold, and phosphorus-deficient stresses. The results revealed that most *OsDUF506s* respond to ABA and JA treatment, as well as drought and cold conditions. In conclusion, our findings provided insights into the evolution and function of *OsDUF506s*, which could benefit crop breeding in the future.

# INTRODUCTION

Domains of unknown functions (DUFs) are batches of gene families with conserved domains but unknown functions that are common in eukaryotes (*Bateman, Coggill & Finn, 2010*). The Pfam database contains 4,716 DUF families (https://www.ebi.ac.uk/interpro/entry/pfam/). Although the majority of DUF families remain unknown, some have been studied. In *Oryza sativa*, *OsDUF1618* (*Wang et al., 2014*), *OsDUF221* (*Ganie, Pani & Mondal, 2017*), *OsDUF1110* (*Harada et al., 2016*), *OsDUF810* (*Li et al., 2018*), OsDUF668 (*Zhong et al., 2019a*), *OsDUF231* (*Zhong, Cui & Ye, 2019b*), *OsDUS936* (*Li et al., 2017*) have been characterized. Previous studies showed that DUF genes were engaged in different

biological functions in *Oryza sativa*. For instance, *SWOLLEN TAPETUM AND STERILITY 1 (STS1)*, which contained a DUF726 domain, interacted with *Polyketide Synthase 2 (OsPKS2)* and *Acyl-CoA Synthetase 12 (OsACOS12)* to contribute to sporopollenin biosynthesis (*Yuan et al., 2022*). Another DUF726 protein, encoded by *Leaked and Delayed Degraded Tapetum 1 (OsLDDT1)*, was involved in fatty acid synthesis and anther epidermis formation (*Sun et al., 2023*). *ROLLED and ERECT LEAF 2* (REL2), which contained DUF630 and DUF632 conserved domains, was involved in regulating leaf morphology, its functional loss resulted in rolling leaves (*Yang et al., 2016*). DUF genes have also been implicated in various biotic and abiotic responses. For example, the *Oryza sativa Stress Responsive DUF740 Protein (OsSRDP)* gene belonged to the DUF740 family, and its overexpressed transgenic plants, driven by promoter AtRd29A, revealed increased resistance to drought, salinity, and cold stresses as well as rice blast fungus (*Jayaraman et al., 2022*). The *DUF966-stress repressive gene 2 (OsDSR2)* gene was involved in the negative regulation of salt and drought stress responses (*Luo et al., 2014*).

The DUF506 family, also called the PDDEXK_6 family, is a group of plant proteins that are distant homologs of the PD-(D/E)XK nuclease superfamily. The nuclear structure is retained as α-β-β-α-β and includes the typical PDDEXK motifs II and III in modified forms as xDxxx motif located in the second core beta-strand, where x is any hydrophobic residue, and a (D/E)X(D/N/S/C/G) pattern. The missing positively charged residue of motif III may be replaced by a conserved arginine in motif IV, which is located in the proceeding alpha-helix (*Knizewski et al., 2007*). So far, DUF506 proteins in *Oryza sativa* have not been characterized systemically and functionally. Previous research merely identified that the expressions of 13 *AtDUF506s* were ubiquitous in organs and associated with abiotic stresses and ABA response (*Ying, 2021*). The comparative microarray data showed that *AT2G20670* was inhibited by *B. cinereal*, heat, salinity, and osmotic stress (*Sham et al., 2019*). Recent studies revealed that *REPRESSOR OF EXCESSIVE ROOT HAIR ELONGATION 1 (AtRXR1)* gene encoded AT3G25240 protein, which was strongly induced by phosphorus limitation and suppressed root hairs (RHs) extension by interacting with RabD2c GTPase. Moreover, its function under phosphorus limitation was conserved in both monocot and dicot, as proven by the analogous function of *Brachypodium distachyon DUF506* (*Ying et al., 2022*). AtRXR3 (AT1G62420), another P-inducible *AtDUF506* gene, inhibited RHs elongation by a distinct mechanism. *AtRXR3* was transactivated by *ROOT HAIR DEFECTIVE6-LIKE4 (RSL4)* and interacted with cytosolic calmodulins to repress RHs elongation (*Ying & Scheible, 2022*). Current studies on DUF506s in Arabidopsis suggested that DUF506 family members were significant in plant growth and abiotic resistance, but DUF506s functions in *Oryza sativa* have rarely been investigated. *LOC_Os01g68650*, the closest homologous gene of *AtRXR1*, was upregulated under drought stress, and its expression in *drought tolerance 11 (OsDT11)* overexpression line was higher than in the wild line, indicating that this gene may participate in drought response and be enhanced by *OsDT11* (*Zhao et al., 2020*). *LOC_Os01g54340* was identified as a nitrogen-sensitive gene that was rapidly repressed by nitrogen starvation (*Hsieh et al., 2018*). Until now, the *OsDUF506* family has not been genome-wide identified and the functions are still unknown.

In this study, we identified all *OsDUF506* family members in *Oryza sativa* by bioinformatic methods. The phylogeny, conserved motifs, cis-acting regulatory elements, distribution of non-synonymous SNPs, target miRNA, synteny, and tissue expression specificity were analyzed. The expression pattern of *OsDUF506s* under plant hormones treatments and drought, cold, and phosphorus-deficient stresses was investigated using transcriptome and qRT-PCR. Our study provided a more comprehensive identification and classification of *OsDUF506s*, broadened our recognition of the functions under abiotic stresses, and laid the groundwork for molecular breeding in *Oryza sativa*.

## MATERIALS AND METHODS

### Identification of *DUF506* members in 10 plant species

All the genome databases were downloaded from the EnsemblPlants database (http://plants.ensembl.org). Thirteen Arabidopsis DUF506 protein sequences were obtained from UniProtKB/Swiss-Prot (SwissProt) database (https://www.uniprot.org) and used as queries to perform protein blast search in 10 plant species (*Oryza sativa*, *Arabidopsis thaliana*, *Glycine max*, *Solanum tuberosum*, *Gossypium raimondii*, *Zea mays*, *Hordeum vulgare*, *Triticum aestivum*, *Sorghum bicolor* and *Ananas comosus*) by using TBtools (*Chen et al., 2020*). Meantime, the typical domain of DUF506 (PF04720, PDDEXK_6) was downloaded from the PFAM database (http://pfam.xfam.org) and was used to search for DUF506 with the HMMER tool (https://www.ebi.ac.uk/Tools/hmmer/search/hmmscan). All the candidates were merged and edited to eliminate redundancies. The candidates were then submitted to NCBI-CDD (https://www.ncbi.nlm.nih.gov/cdd/) to test for the existence of the complete DUF506 conserved domain and those with an incomplete N end or C end were eliminated. The molecular weight (Mw), isoelectric point (pI), instability index, aliphatic index, and grand average of hydropathicity (GRAVY) of DUF506 members were predicted with ExPASy (http://web.expasy.org/protparam). The subcellular localizations were predicted with WoLF PSORT (https://wolfpsort.hgc.jp).

### Phylogenetic relationship, structure, and conserved motifs analysis of *OsDUF506* members

A total of 130 *DUF506s* was used for aligning with the MUSCLE method (default parameters) and construction of an ML phylogenetic tree (default parameters) using MEGA 11 software by setting bootstrap to 1,000 and JTT+G model (*Tamura, Stecher & Kumar, 2021*). The result was displayed by ChiPlot (https://www.chiplot.online). The conserved motifs were predicted with MEME (https://meme-suite.org/meme/doc/meme.html) and visualized by TBtools (*Chen et al., 2020*).

### Prediction of CREs of *OsDUF506* members

The 2,000 bp upstream sequences of promoters were used to search for CREs with PlantCARE (http://bioinformatics.psb.ugent.be/webtools/plantcare/html) and visualized by TBtools (*Chen et al., 2020*).
## Analysis of SNPs of *OsDUF506* members

The *OsDUF506* sequences were used to query the SNP-Seek database for nsSNPs against Nipponbare reference (https://snp-seek.irri.org). SNP-index was used to assess the subpopulation specificity of nsSNPs. The SNP-index was calculated as $SNP_{GJ}\text{-index} = N_{ref}/N_{all} \times 100\%$, $SNP_{H}\text{-index} = N_{h}/N_{all} \times 100\%$, $SNP_{XI}\text{-index} = 1\text{-}SNP_{GJ}\text{-index-}SNP_{H}\text{-index}$, $N_{ref}$ represented the number of varieties sharing the same allele with reference, $N_{h}$ represented the number of varieties with heterozygous allele, $N_{all}$ represented the total number of varieties with determined alleles at the SNP locus.

## Syntenic analysis of *OsDUF506* members

The duplication events and syntenic relationship of *DUF506s* between *Oryza sativa* and other plants were obtained by MCScanX (*Wang et al., 2012*). The results were visualized by TBtools (*Chen et al., 2020*). The nonsynonymous (Ka) and synonymous (Ks) calculations were performed by the simple Ka/Ks calculator kit of TBtools (*Chen et al., 2020*).

## Predict analysis of miRNAs interacting with *OsDUF506* members

The miRNAs targeting *OsDUF506s* were predicted by psRNATarget (https://www.zhaolab.org/psRNATarget/analysis?function=2) and visualized by ChiPlot (https://www.chiplot.online). The expressions of the predicted miRNAs were obtained from the PmiRExAt database (http://pmirexat.nabi.res.in/searchdb.html) and the normalized TPM values (log2) were used for constructing a heatmap, the scale method of normalized was used to intuitively reflect the expressing differences between tissues and treatments by TBtools (*Chen et al., 2020*).

## Expression analysis of *OsDUF506* members in different tissues and induced by plant hormones

The expression data in various tissues and induced by 50 μM abscisic acid (ABA), 10 μM gibberellin 3 (GA$_3$), 10 μM auxin (IAA), 100 μM jasmonic acid (JA), 1 μM brassinolide (BL), and 1μM trans-Zeatin (tZ) were obtained from RiceXPro database (https://ricexpro.dna.affrc.go.jp/quick-guide.html), the normalized signal intensity values (log$_2$) were used for constructing heatmap, and the scale method of normalized was used to intuitively reflect the expressing changes of particular genes at different treating time points by TBtools (*Chen et al., 2020*).

## Plant growth conditions and abiotic stresses treatments

Japonica rice variety Yunkegeng 5 was used in the expression analysis. The plants were cultivated in a climate chamber (160 μmol$^{-2}$ s$^{-1}$ light intensity, 14 h-light and 10 h-dark a cycle, 28 °C, 45% RH) for 14 days in Yoshida rice nutrient solution (NS1040; Coolaber Technology Co., Ltd., Beijing, China). For drought stress, plants were transferred to a nutrient solution of 20% (w/v) polyethylene glycol (PEG-6000) for 3 h. For cold stress, plants were transferred to the climate chamber under 4 °C treatment for 3 h.
For phosphorus deficiency stress, plants were transferred to phosphorus-deficient Yoshida nutrient solution (NSP1040-P; Coolaber Technology Co., Ltd., Beijing, China) for 7 days.
Three biological replicates of the treated and control plants were harvested and stored at −80 °C.

## Expression analysis of *OsDUF506* members by qRT-PCR

Primer design and specificity check was performed by Primer-BLAST of NCBI (https://www.ncbi.nlm.nih.gov/tools/primer-blast/index.cgi). The total RNA was extracted by TaKaRa MiniBEST Plant RNA Extraction Kit, cDNA was synthesized using Vazyme HiSript III 1st Strand cDNA Synthesis Kit (+gDNA wiper). The qRT-PCR was accomplished by Vazyme ChamQ SYBR Color qPCR Master Mix (Without ROX) in LightCycler96 system under the PCR condition of 95 °C for 60 s, 45 cycles of 95 °C for 10 s, 54 to 60 °C for 20 s and 72 °C for 20 s. The relative expressions data were calculated by $2^{-\Delta\Delta CT}$ method, and the reference used in this study was *OsActin*. The primer sequences were listed in Table S9. The significant differences level was analyzed by unpaired t-test with GraphPad Prism 8.0 software.

# RESULTS

## Identification and phylogenetic relationship of *DUF506* members in *Oryza sativa* and nine additional plant species

Using the previous method, we identified 10 *DUF506*s in *Oryza sativa* and named them *OsDUF50601* to *OsDUF50610*. Moreover, 13, 25, 13, 20, 8, 4, 22, 8, 7 *DUF506s* were identified in *Arabidopsis thaliana*, *Glycine max*, *Solanum tuberosum*, *Gossypium raimondii*, *Zea mays*, *Hordeum vulgare*, *Triticum aestivum*, *Sorghum bicolor* and *Ananas comosus*, respectively (Table S1). A total of 130 *DUF506s* were used to construct an ML phylogenetic tree and divided into four subfamilies based on the evolutionary distance referred to previous literature (*Ying, 2021*). The IIIb subfamily contained the most *DUF506s*, while the IIIa subfamily had the fewest. Three members belonged to subfamily I (*OsDUF50602*, *OsDUF50609*, and *OsDUF50610*), two belonged to subfamily II (*OsDUF50601* and *OsDUF50606*), only *OsDUF50603* belonged to subfamily IIIa, while the rest four members composed the largest subfamily IIIb (Fig. 1).

## Characterization and conserved motifs of Os*DUF506* members

The 10 *OsDUF506s* were located on chromosome 1, 3, 5, 7, 10, 11. The summary of characteristics was shown in Table 1. Their protein lengths ranged from 269 to 507. The molecular weight, theoretical isoelectric points, and aliphatic index were predicted between 29,861.74 to 54,428.55 Da, 5.58 to 9.12, and 62.39 to 86.85, respectively. The instability index of proteins exceeded 40, and their GRAVY values were negative, suggesting that they were unstable hydrophilic proteins. The subcellular localization of subfamily II members (*OsDUF50601* and *OsDUF50606*) was predicted to be in the nucleus, while the rest were predicted to be in the chloroplast. *OsDUF506s* possessed 1 to 3 exons, and the exon/intron structure of genes within a subfamily was comparable (Fig. 2A). The *OsDUF506s* from subfamily I had only one CDS region, while those from subfamilies II and IIIa had three CDS regions. The number of CDS regions led to the division of
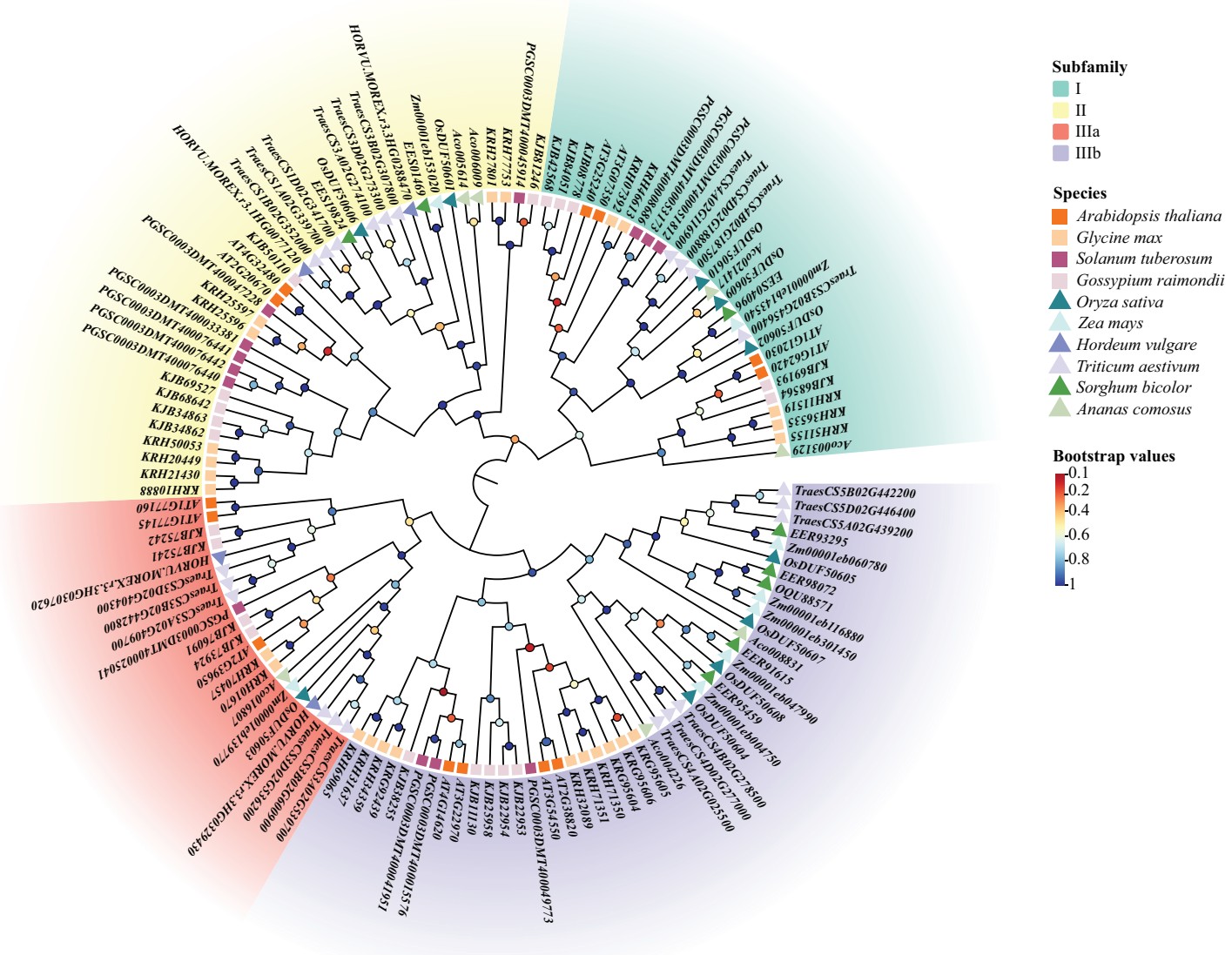

**Figure 1 Phylogenetic tree of *DUF506* members identified in ten plant species.** The members were divided into four subfamilies shown in different colors. Circles in different colors represent different bootstrap values. Bootstrap values were calculated from 1,000 replications and only the values over 0.7 bootstrapping were considered significant.

subfamily IIIb into two branches. The result suggested that members of different subfamilies might have distinct functions.

Fifteen conserved protein motifs of *OsUDF506s* were identified (Fig. 2B). All *Oryza sativa* and Arabidopsis *DUF506* members shared motifs 1, 2, 3, and 5. In subfamily I, the most conserved motif was 4 and 11. Ultimately, *OsDUF50602* and *OsDUF50610* shared the same motifs with *AT1G62420* and *AT3G25240*, indicating they may have the same biological function. The subfamily II members shared motifs 8 and 11. The motif construction of *OsDUF50602* from subfamily IIIa was identical to that of *AT2G39650*. All subfamily IIIb members shared motifs 4, 6, and 7. Three-quarters possessed motif 9,
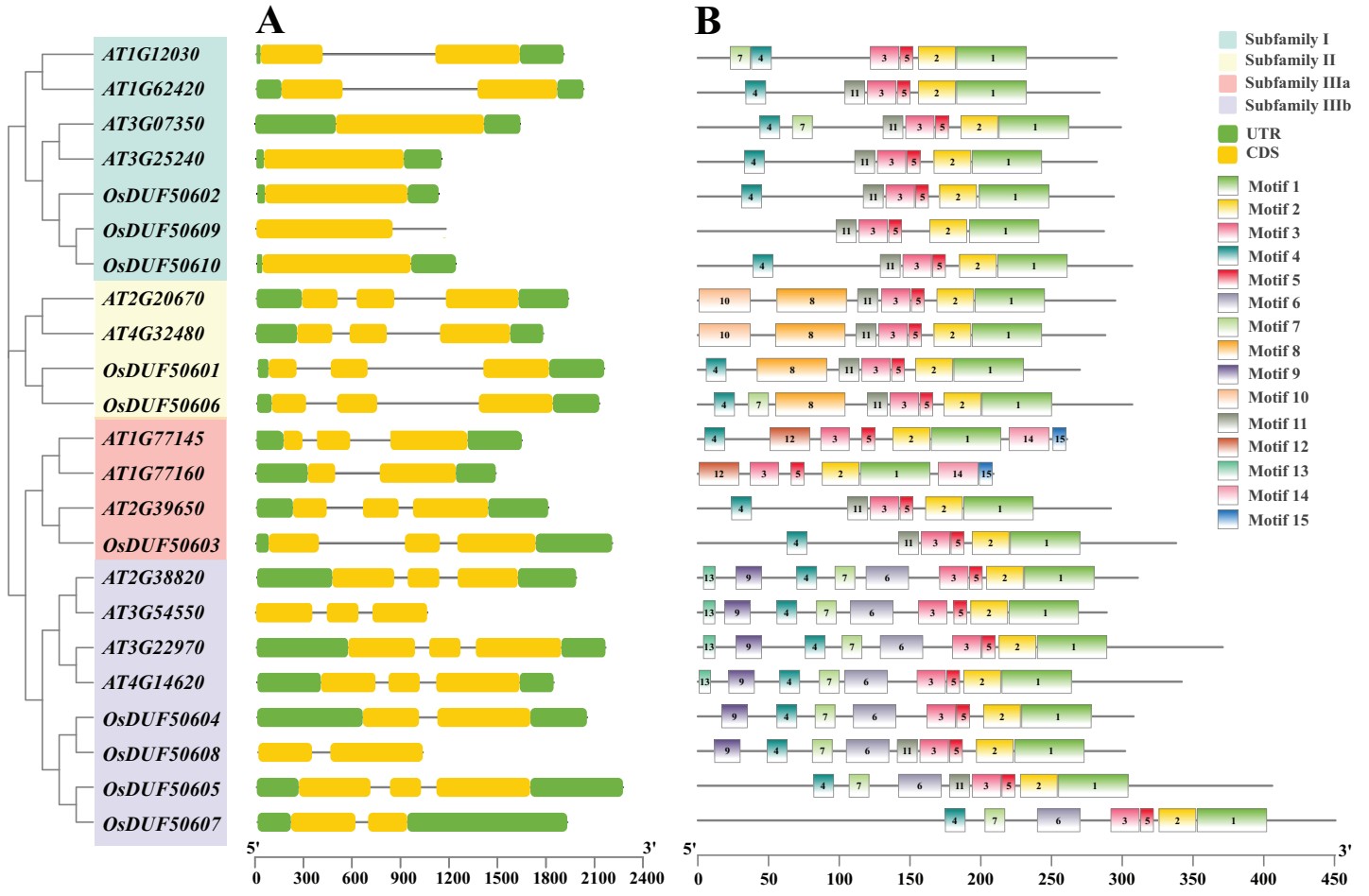

**Figure 2 Gene structure, and conserved motifs of *DUF506* in *Oryza sativa* and *Arabidopsis thaliana*.** Four background colors indicate four subfamilies. (A) Exon/intron structures. Green bars, yellow bars, and lines indicate UTRs, exons, and introns, respectively. (B) Distributions of conserved motifs. Motifs are shown by 15 different color bars.

but motif 13 was exclusive to genes from Arabidopsis. The result showed that *DUF506s* belonging to the same subfamily shared similar motif characteristics, indicating that they have a similar function.

## Analysis of cis-acting regulatory elements (CREs) of *OsDUF506* members

To analyze the prospective function of *OsDUF506s*, we searched the promoter regions (2,000 bp upstream of the start codon) and predicted 321 potential CREs (Fig. 3). *OsDUF50604* possessed the most CREs, while *OsDUF50603* and *OsDUF50609* possessed the fewest (Table S2). Nineteen types of CREs were identified and grouped into three functional categories: hormone response, abiotic stress response, and plant growth and metabolism. Hormone response-related CREs included MeJA-response elements (TGACG-motif and CGTCA-motif), abscisic acid response elements (ABRE), auxin response elements (TGA and AuxRR-core), salicylic acid response elements (TCA) and gibberellin response elements (GARE-motif and P-box). Each member of *OsDUF506s*

**Table 1 Characteristics of *DUF506* genes in *Oryza sativa*.**

| Gene ID | Subfamily | Locus ID | Size (aa) | Protein MW (Da) | pI | Instability index | Aliphatic index | GRAVY | Chromosome | Subcellular localization |
|---------|-----------|----------|-----------|-----------------|-----|-------------------|-----------------|-------|------------|--------------------------|
| *OsDUF50601* | II | LOC_Os01g54340 | 269 | 29,861.74 | 7.61 | 47.02 | 65.76 | −0.460 | 1 | Nucleus |
| *OsDUF50602* | I | LOC_Os01g68650 | 293 | 30,429.34 | 8.97 | 52.76 | 75.22 | −0.134 | 1 | Chloroplast |
| *OsDUF50603* | IIIa | LOC_Os01g74250 | 337 | 37,616.96 | 7.25 | 60.92 | 86.85 | −0.325 | 1 | Chloroplast |
| *OsDUF50604* | IIIb | LOC_Os03g06680 | 307 | 33,011.21 | 6.68 | 45.12 | 80.59 | −0.416 | 3 | Chloroplast |
| *OsDUF50605* | IIIb | LOC_Os03g58230 | 405 | 43,737.47 | 9.12 | 50.54 | 68.52 | −0.489 | 3 | Chloroplast |
| *OsDUF50606* | II | LOC_Os05g44300 | 306 | 32,169.30 | 6.76 | 56.42 | 62.39 | −0.267 | 5 | Nucleus |
| *OsDUF50607* | IIIb | LOC_Os07g08390 | 507 | 54,428.55 | 9.07 | 59.08 | 66.07 | −0.481 | 7 | Chloroplast |
| *OsDUF50608* | IIIb | LOC_Os10g28210 | 301 | 32,041.06 | 6.52 | 50.73 | 79.60 | −0.347 | 10 | Chloroplast |
| *OsDUF50609* | I | LOC_Os11g25020 | 286 | 30,211.19 | 9.03 | 42.62 | 76.71 | −0.108 | 11 | Chloroplast |
| *OsDUF50610* | I | LOC_Os11g25040 | 306 | 32,464.54 | 5.58 | 50.32 | 82.06 | −0.176 | 11 | Chloroplast |

contained at least two types of hormone response elements, MeJA-response elements and abscisic acid response elements. Regarding abiotic stress response, light response elements, anaerobic induction elements, and anoxic specific inducible elements were the most prevalent. In subfamily II and IIIb, there were low-temperature response (LTR) elements. Drought inducible elements (MSB) existed in *OsDUF50601*, *OsDUF50602*, *OsDUF50607*, and *OsDUF50609*, indicating that they may be associated with drought stress. Fewer CREs were related to plant growth and metabolism. CAT-boxes relating to meristem expression were found in half of *OsDUF506s*, while Motif I in *OsDUF50601*, *OsDUF50605*, and *OsDUF50606* was involved in root specificity. Only a few *OsDUF506* members possessed CREs associated with circadian control, cell cycle regulation, zein metabolism regulation, endosperm expression, and flavonoid biosynthetic genes regulation. These results suggested that *OsDUF506s* might be essential in hormone regulation and abiotic stress response.

## Analysis of SNPs in *OsDUF506* members

According to the Rice SNP-Seek Database, 78 SNPs were identified in *OsDUF506s*, of which 30 were non-synonymous SNP (nsSNP) (Table 2). *OsDUF50609* and *OsDUF50610* possessed 13 and six nsSNPs, respectively, while the other members possessed between one to three nsSNP, indicating they were relatively conserved. To explore the subpopulation-specific variants, 30 nsSNP genotyping data of 2,644 *Oryza sativa* varieties from nine subpopulations (five *Xian/indicia (XI)* subpopulations and four *Geng/japonica (GJ)* subpopulations) were analyzed. SNP$_{GJ}$-index represented the proportion of varieties that shared the same allele as the reference Nipponbare. Three nsSNPs (OsDUF50609.1, OsDUF50609.7, and OsDUF50610.6) from subfamily I and one nsSNP (OsDUF50607.1) from subfamily IIIb were specific between Indica and Japonica varieties, because the SNP$_{GJ}$-index values in Indica subpopulations were less than 5%, while the values in the Japonica subpopulations were greater than 85% (Fig. 4 and Table S3).

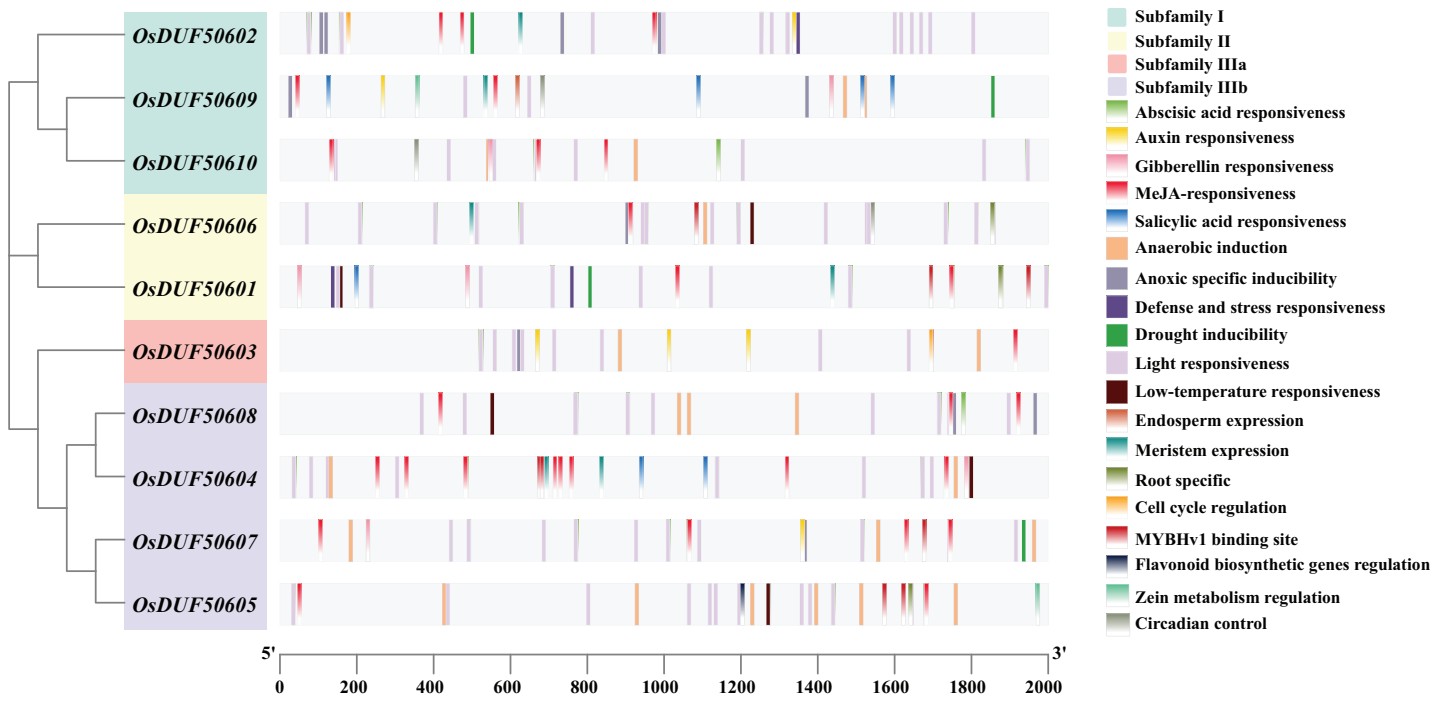

**Figure 3 Distribution of CREs in *OsDUF506s* promoters.** Predicted CREs with different functions are displayed with boxes in different colors. The light gray bars represent the 2,000 bp sequence upstream of *OsDUF506s*.

## Synteny analysis of *OsDUF506* members

Gene duplications are significant for gene family evolution. The synteny analysis result showed three segmental duplications (*OsDUF50601-OsDUF50606*, *OsDUF50604-OsDUF50608*, and *OsDUF50605-OsDUF50607*) and one tandem duplication (*OsDUF50609-OsDUF50610*) in *Oryza sativa* (Fig. 5), indicating that segmental duplication was the core expansion dynamic of *OsDUF506s* evolution. All duplications had Ka/Ks ratios below 0.5 (Table S4), demonstrating that *OsDUF506s* have undergone purifying selective pressure during evolution. In addition, these duplicated events were speculated to occurred at least 20.48 million years ago (Table S4).

Besides, duplicated events of *DUF506s* in *Oryza sativa*, the dicot model plant (*Arabidopsis thaliana*), and other monocotyledonous species (*Zea mays*, *Hordeum vulgare*, *Triticum aestivum*, *Sorghum bicolor*, and *Ananas comosus*) were also identified. *OsDUF50604*, *OsDUF50605*, *OsDUF50607*, and *OsDUF50608* in *Oryza sativa* and *AT3G22970* and *AT4G14620* in *Arabidopsis* from subfamily IIIb formed six homologous gene pairs and showed multiple collinearities (Fig. 5). In five monocotyledonous species, other than *OsDUF50609* and *OsDUF50610* on chromosome 11, the other *OsDUF506s* all had homologous genes. A total of 10 colinear gene pairs were found between *Oryza sativa* and *Hordeum vulgare*, 11 pairs with *Zea mays* and *Sorghum bicolor*, respectively, 12 pairs with *Ananas comosus*, and 30 pairs with *Triticum aestivum* (Fig. 6 and Table S5). The results suggested that these *DUF506s* might be derived from the same ancestral type

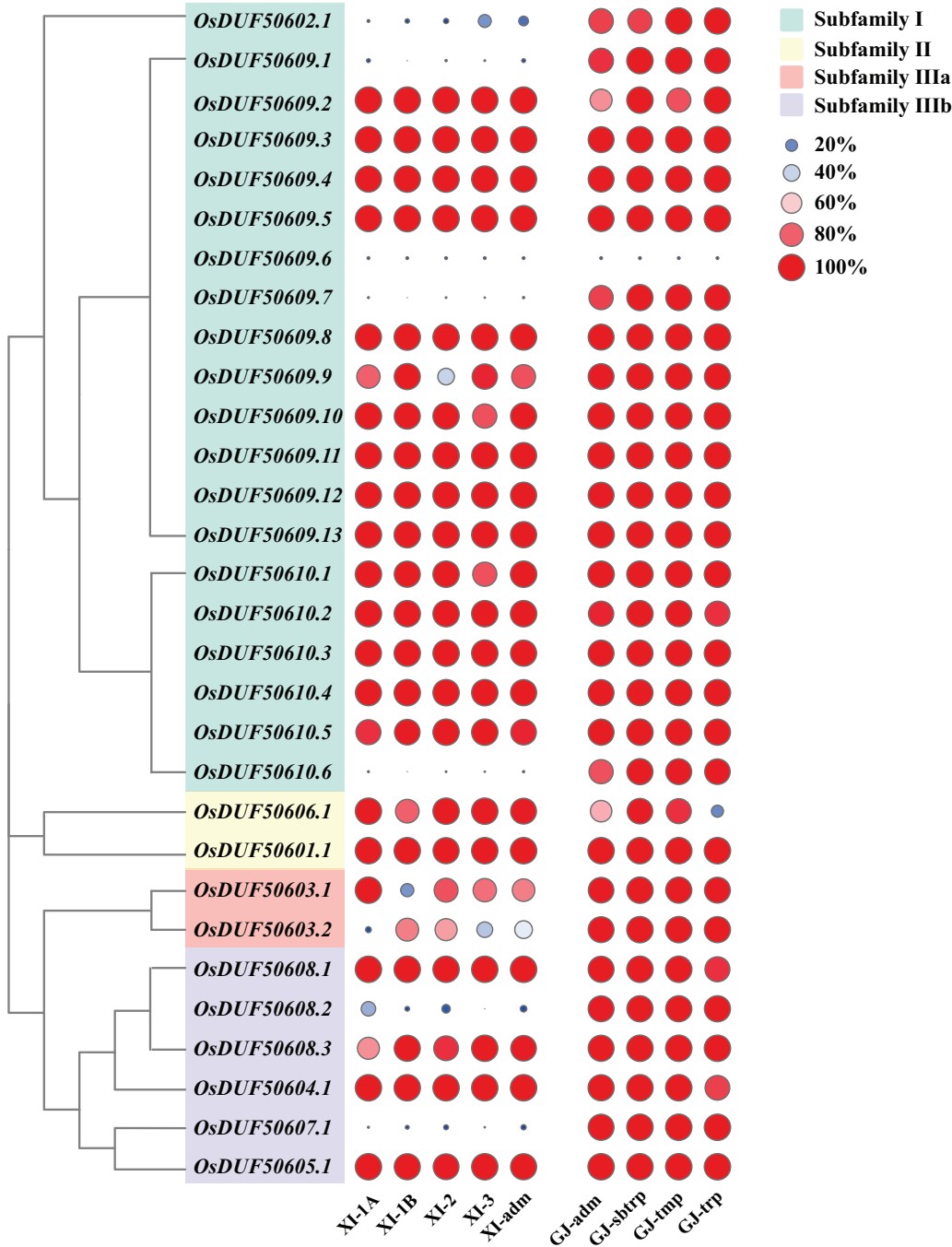

**Figure 4 SNP$_{GJ}$-index values of *OsDUF506s* nsSNPs in nine subpopulations.** Circles in different colors and sizes represent different SNP$_{GJ}$-index values.

and function similarly. The *OsDUF506s* from subfamily IIIb were involved in gene pairs with *Arabidopsis thaliana and* those five monocots, respectively, indicating that they might be critical to the evolution of the DUF506 family.

**Table 2  Non-synonymous SNP distribution of *OsDUF506s* in 3,024 *Oryza sativa* varieties.**

| Subfamily | Gene ID | No. total SNP | nsSNP ID | Ref allele | Alt allele | Position |
|---|---|---|---|---|---|---|
| I | *OSDUF50602* | 2 | OSDUF50602.1 | C | T | Chr1:39867256 |
| | *OsDUF50609* | 23 | OsDUF50609.1 | G | T | Chr11:14248387 |
| | | | OsDUF50609.2 | C | A | Chr11:14248411 |
| | | | OsDUF50609.3 | C | T | Chr11:14248432 |
| | | | OsDUF50609.4 | A | G | Chr11:14248446 |
| | | | OsDUF50609.5 | C | T | Chr11:14248498 |
| | | | OsDUF50609.6 | A | G | Chr11:14248539 |
| | | | OsDUF50609.7 | C | G | Chr11:14248616 |
| | | | OsDUF50609.8 | C | G | Chr11:14248636 |
| | | | OsDUF50609.9 | C | T | Chr11:14248665 |
| | | | OsDUF50609.10 | G | T | Chr11:14248737 |
| | | | OsDUF50609.11 | C | T | Chr11:14248861 |
| | | | OsDUF50609.12 | A | C | Chr11:14248932 |
| | | | OsDUF50609.13 | T | C | Chr11:14249448 |
| | *OsDUF50610* | 9 | OsDUF50610.1 | G | T | Chr11:14267840 |
| | | | OsDUF50610.2 | C | T | Chr11:14267960 |
| | | | OsDUF50610.3 | G | A | Chr11:14268083 |
| | | | OsDUF50610.4 | C | G | Chr11:14268298 |
| | | | OsDUF50610.5 | C | T | Chr11:14268593 |
| | | | OsDUF50610.6 | T | C | Chr11:14268614 |
| II | *OsDUF50606* | 3 | OsDUF50606.1 | G | A | Chr5:25778321 |
| | *OSDUF50601* | 9 | OSDUF50601.1 | A | G | Chr1:31272839 |
| IIIa | *OSDUF50603* | 12 | OSDUF50603.1 | G | A | Chr1:43017098 |
| | | | OSDUF50603.2 | C | A | Chr1:43018592 |
| IIIb | *OsDUF50608* | 7 | OsDUF50608.1 | G | C | Chr10:14658953 |
| | | | OsDUF50608.2 | G | A | Chr10:14658996 |
| | | | OsDUF50608.3 | C | G | Chr10:14659216 |
| | *OsDUF50604* | 3 | OsDUF50604.1 | T | C | Chr3:3381185 |
| | *OsDUF50607* | 3 | OsDUF50607.1 | G | C | Chr7:4305870 |
| | *OsDUF50605* | 7 | OsDUF50605.1 | T | A | Chr3:33171316 |

## Predicted miRNAs analysis

Sixty-nine predicted miRNAs that target *OsDUF506s* and might be involved in expression regulations were identified (Fig. 7A). All *OsDUF506s* were targeted by multiple miRNAs, suggesting these genes were strictly regulated by the combination of multiple miRNAs. Multiple *OsDUF506s* were targeted by *osa-miRNA2927, osa-miRNA5075, osa-miRNA1848, osa-miRNA2925,* and *osa-miRNA5809,* indicating that these miRNAs were essential to *OsDUF506s*. The lengths of matured miRNAs ranged between 19 and 24 nucleotides (Table S6). Most of the miRNAs inhibited *OsDUF506* expressions by cleavage, only *osa-miR2880, osa-miR5340, osa-miR2926, osa-miR156j-3p, osa-miR1875, osa-miR444b.1, osa-miR444c.1,* and *osa-miR5832* inhibited *OsDUF506* expressions by translation repression.

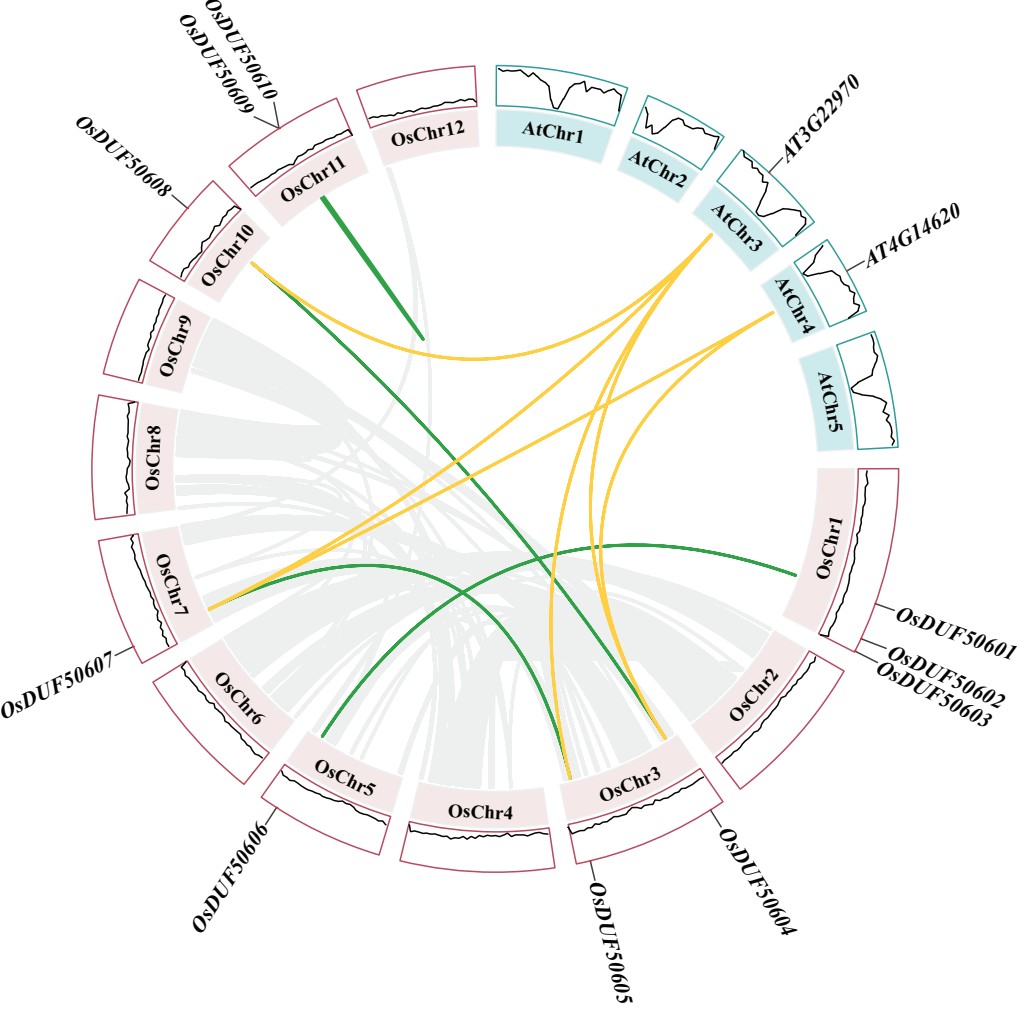

**Figure 5  Synteny analysis of *DUF506*s in *Oryza sativa* and *Arabidopsis*.** Pink and green bars represent the chromosomes of *Oryza sativa* and *Arabidopsis* respectively. The black lines in the colored box show the gene density of the chromosomes. The green lines suggest duplicated gene pairs in *Oryza sativa*, the yellow lines indicate the collinearity between *Oryza sativa* and *Arabidopsis thaliana*.

The expression analysis revealed that the majority of miRNAs were expressed at a modest level in *Oryza sativa* tissues (Fig. 7B). The osa-miR1874-3p, which targeted *OsDUF50605* specifically, showed the highest expression level in embryo. The osa-miR444b.1 targeting *OsDUF50606* was highly expressed in all tissues except anther, indicating that *OsDUF50606* expression was limited in these tissues, which may inhibit the function of *OsDUF50606* in plant growth. The expression levels of miRNAs under drought, salt, and cold stress showed no significant changes, suggesting they did not participate in abiotic stress responses.

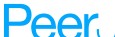

**Figure 6 Synteny analysis of *DUF506s* between *Oryza sativa* and five monocot species.** Red lines suggest syntenic gene pairs.

## Expression analysis of *OsDUF506* members in different tissues and induced by plant hormones

To investigate the expression specificities of *OsDUF506s*, the expression values in the leaf blade, leaf sheath, root, stem, inflorescence, anther, pistil, lemma, palea, ovary, embryo, and endosperm were analyzed (Fig. 8; Table S7). *OsDUF50602* from subfamily I expressed

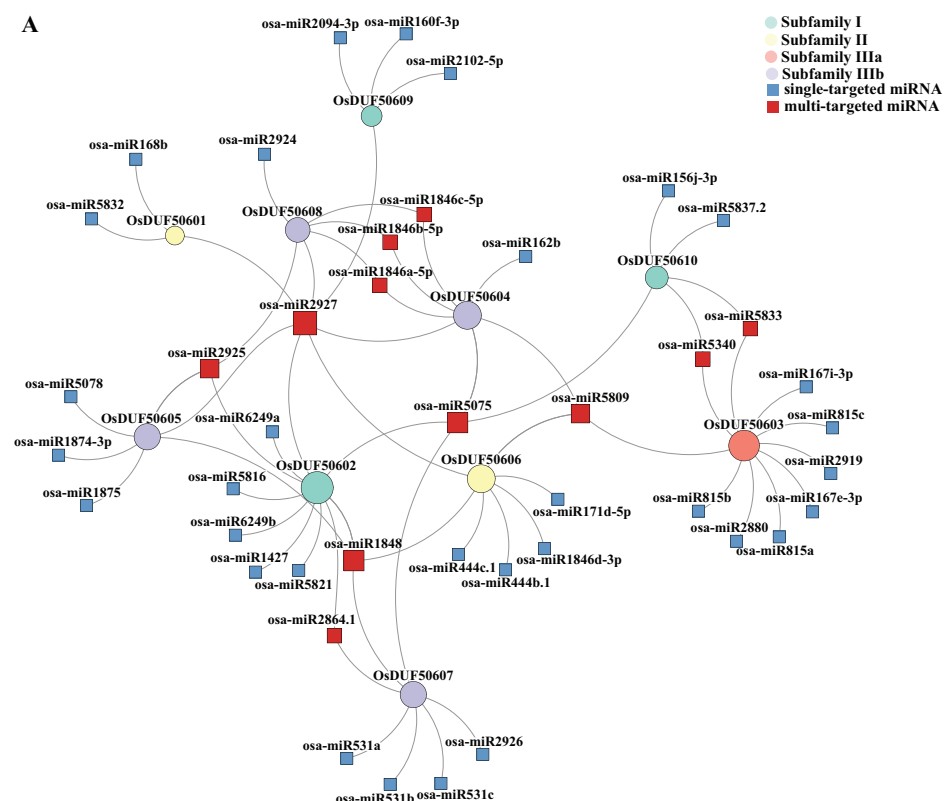

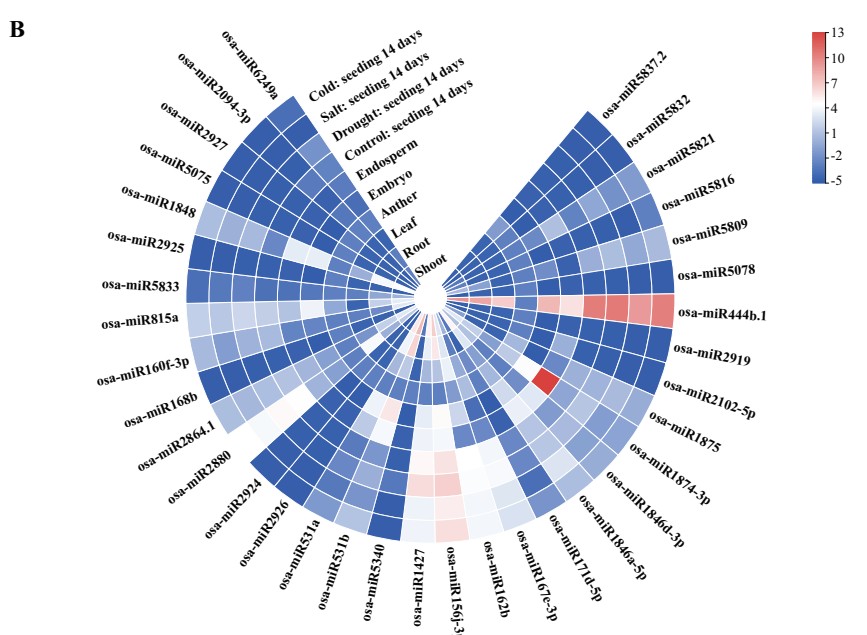

**Figure 7 Analysis of predicted miRNA targeting *OsDUF506s*.** (A) Identified miRNAs targeting *OsDUF506s*. Circles represent *OsDUF506s*, squares represent the related miRNAs. (B) Expressions of predicted miRNAs in different tissues and under abiotic stresses. The heatmap demonstrates the expression level, the color gradient from blue to red presents increasing expression values.

in all tissues, with the highest expressions in the embryo and endosperm 7 days after flowering, whereas *OsDUF506010* was expressed at an exceedingly low level in all tissues. The expression modes of the segment duplication gene pair *OsDUF50601-OsDUF50606* from subfamily II were wholly dissimilar, suggesting that they might be involved in functional redundancy. Compared to the genes from subfamily IIIa, the expression levels of four genes from subfamily IIIb were higher in most tissues. The segment duplication gene pair *OsDUF50604-OsDUF50608* was highly expressed in anther and leaf sheath, respectively. In contrast, another segment pair, *OsDUF50605-OsDUF50607*, was highly expressed in the leaf blade and leaf sheath, revealing that they might be involved in functional differentiation.

The CREs analysis of *OsDUF506s* suggested that they were widely involved in hormonal regulation, thus we analyzed their expression profiles in root and shoot treated with six plant hormones, including abscisic acid (ABA), gibberellic acid (GA$_3$), indole-3-acetic acid (IAA), jasmonic acid (JA), brassinolide (BL), and trans-zeatin (tZ). *OsDUF506s* showed distinct regulating modes (Fig. 9 and Table S8). Significant upregulation of *OsDUF50602* was induced by ABA and JA, whereas opposite regulation was induced by tZ in root and shoot. Subfamily II members, *OsDUF50601* and *OsDUF50606* exhibited identical regulation under ABA treatment. In the root, except *OsDUF50605* was upregulated by ABA induction, the other members from subfamily IIIb were downregulated by ABA, IAA, and JA. Under BL treatment, *OsDUF50604* and *OsDUF50608* were upregulated, whereas *OsDUF50605* and *OsDUF50607* were downregulated in root. Interestingly, *OsDUF50607* was significantly upregulated by ABA induction in shoot, while the opposite was observed in root. *OsDUF50605* also showed opposite effects on shoot and root regulations by ABA induction.

## Expression analysis of *OsDUF506* members under drought stress, cold stress, and phosphorus-deficient stress

To explore the responses of *OsDUF506s* under abiotic stresses, the relative expressions of eight *OsDUF506s* under drought, cold, and phosphorus-deficient stresses were analyzed by qRT-PCR (Fig. 10). Under drought condition, the expressions of *OsDUF50601*, *OsDUF50603*, *OsDUF50604*, *OsDUF50607*, and *OsDUF50608* were significantly downregulated, whereas only *OsDUF50602* was significantly upregulated with a more than 2-fold increase. By contrast, *OsDUF506s* were more sensitive to cold stress. Under cold treatment, six *OsDUF506s* were upregulated with the exception of *OsDUF50606* and *OsDUF50607*, *OsDUF50601*, and *OsDUF50504*, which showed a 4-fold increase in gene expression. Under phosphorus-deficient condition within a week, *OsDUF50601*, *OsDUF50604*, and *OsDUF50607* were significantly downregulated, whereas only *OsDUF50605* was significantly upregulated but by less than 2-fold. The result demonstrated the majority of *OsDUF506s* were induced by drought and cold stresses, suggesting these genes might be implicated in these stress responses.
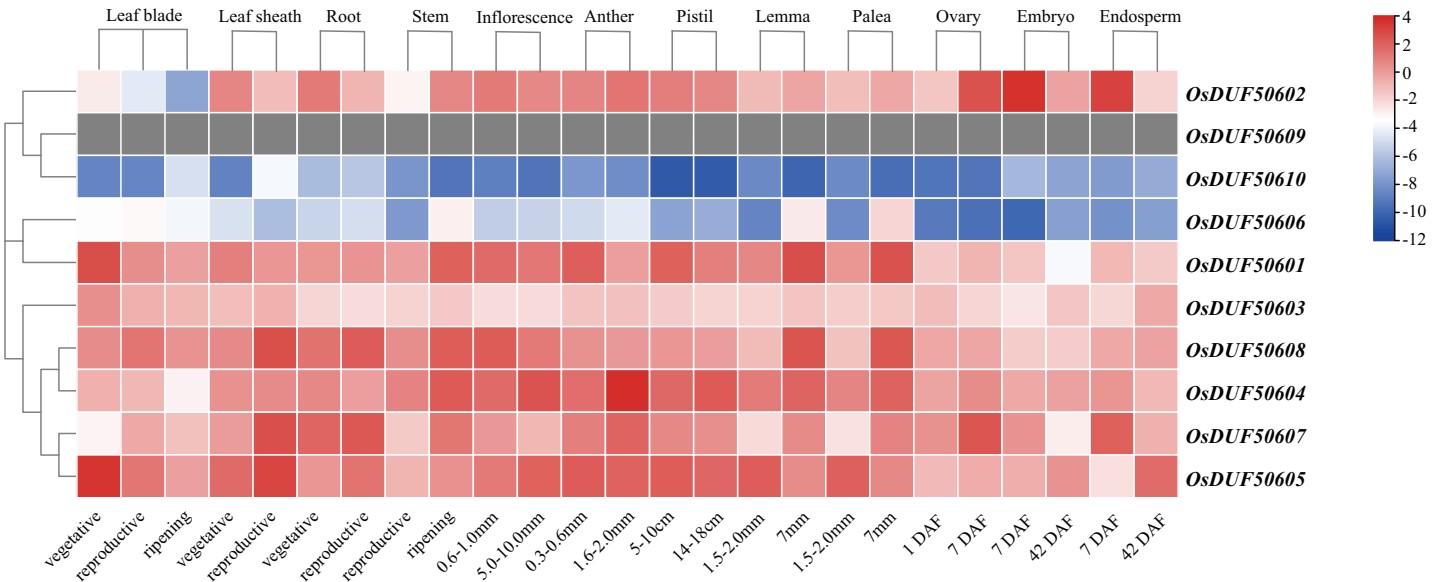

**Figure 8 Expressions of *OsDUF506* members in different tissues.** The heatmap demonstrates the expression level, the color gradient from blue to red presents increasing expression values. Grey presents missing data.

## DISCUSSION

DUFs are a specific type of genes with conserved domains but unknown functions. DUF506 family belongs to the PD-(D/E)XK nuclease superfamily, which consists of numerous enzymes involved in significant cellular processes (*Knizewski et al., 2007*). Most families are certified to be distinct restriction endonucleases with functions including repair of damaged DNA, resolution of holliday junctions, and excessive cleaving in DNA recombination (*Bujnicki, 2003*). The family retains the characteristic motif II of PD-(D/E) XK, but lacks a functionally-characterized domain (*Knizewski et al., 2007*). Until now, *DUF506* family research has only performed in *Arabidopsis* (*Ying, 2021*). In this study, we identified 10 *OsDUF506s* with an intact DUF506 domain in *Oryza sativa* and categorized them into four subfamilies based on the previous study in *Arabidopsis* (Table 1). Additionally, 120 *DUF506* genes from five other monocots and four dicots were also identified and their phylogenetic relationship with *OsDUF506s* was analyzed. In contrast to *DUF1618s*, which only existed in monocot, *DUF506s* were present in both monocots and dicots (Fig. 1), indicating that *DUF506* was an ancient gene family that originated prior to the dicotyledon-monocotyledon divergence (*Wang et al., 2014*). *DUF506* members from monocots or dicots preferred to congregate on the same branch (Fig. 1), indicating a substantial divergence in *DUF506s* between monocots and dicots. Moreover, genome size did not correlate with the number of *DUF506* genes. In dicots, for example, *Arabidopsis thaliana* and *Solanum tuberosum* possessed the same quantity of *DUF506* genes, but their genome sizes differed significantly.

Gene duplication is widespread in plants and contributes to genome expansion and the evolution of new functions. The majority of gene duplications consist of whole-genome duplications and tandem duplications (*Panchy, Lehti-Shiu & Shiu, 2016*). However, in

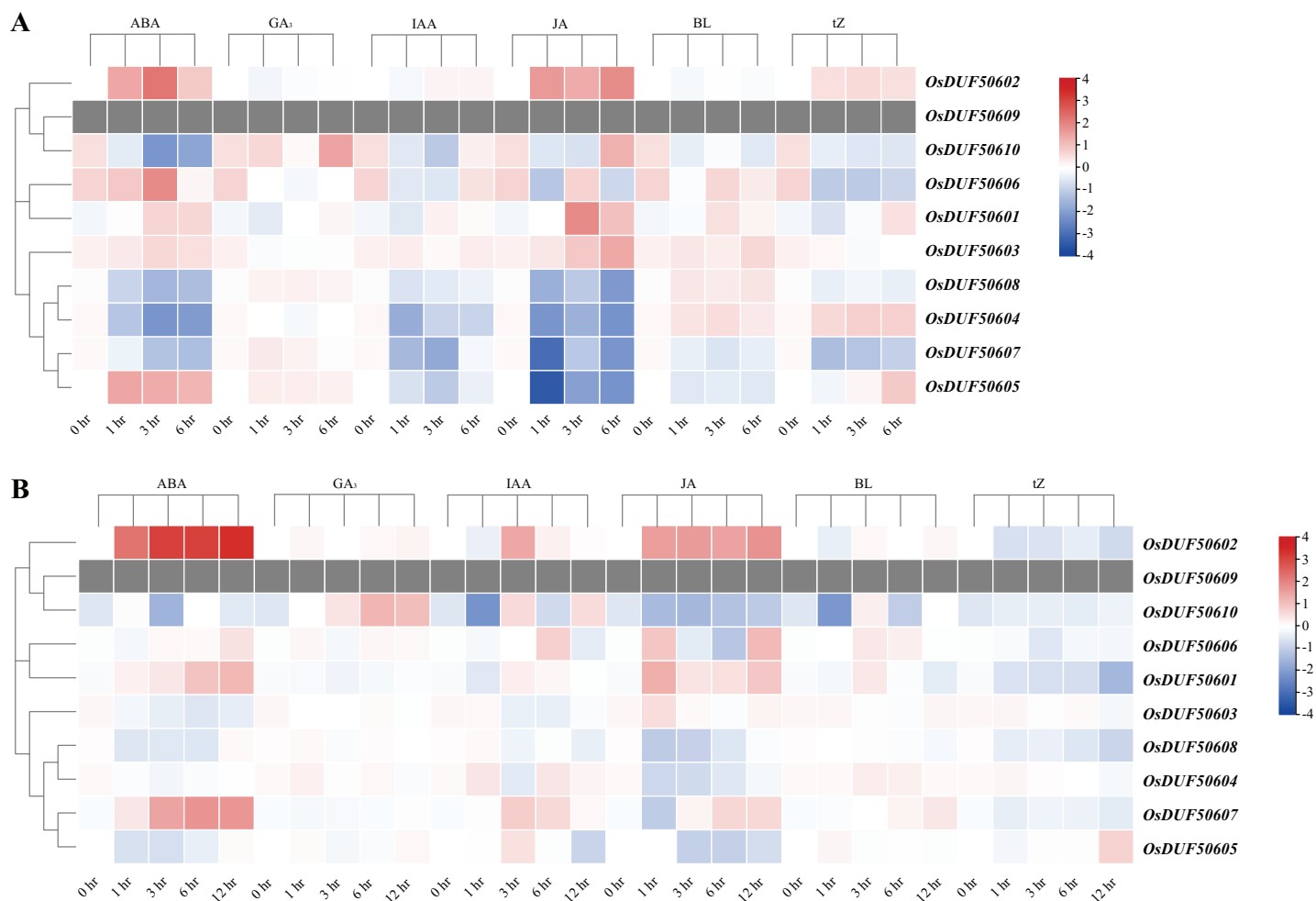

**Figure 9 Expressions of *OsDUF506s* induced by plant hormones in root (A) and shoot (B).** The heatmap demonstrates the expression levels, the color gradient from blue to red presents increasing expression values. Grey presents missing data.

*OsDUF506* duplication events, segmental duplication accounted for 75%, while tandem duplication only accounted for 25% (Table S4), suggesting that segmental duplication was critical for expanding *OsDUF506* family members. The dynamic processes between gene duplications and gene losses contribute to genomes differences (*Holland et al., 2017*). Research on *Oryza sativa* duplications revealed that 85% of duplicates underwent loss, subfunctionalization, or neofunctionalization during 50-70 million years of evolution (*Throude et al., 2009*). *OsDUF50602* and *OsDUF50603* did not form duplicate gene pairs (Fig. 5), indicating they could undergo gene losses. The mechanisms of duplicate gene loss include the deletion of duplicate sequence and pseudogenization, the latter of which refers to gene silence and genetic redundancy (*Ho-Huu et al., 2012*; *Lynch & Conery, 2000*; *Thibaud-Nissen, Ouyang & Buell, 2009*). The duplicated gene with low expression may experience pseudogenization, and pseudogenes typically originate from tandem duplications (*Yang et al., 2011*). This viewpoint is supported by the tandem duplicate pair *OsDUF50609-OsDUF50610*, which originated 58.69 million years ago. According to the

**A**

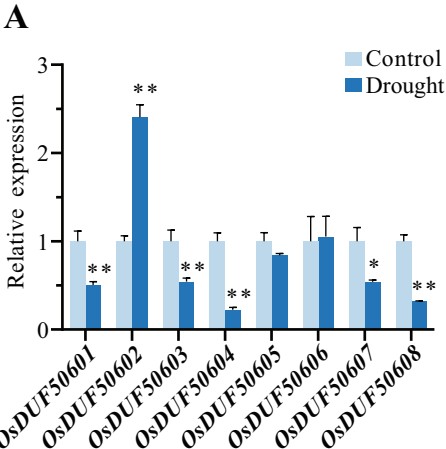

**B**

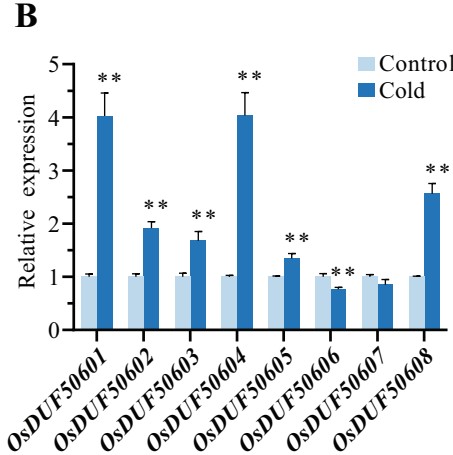

**C**

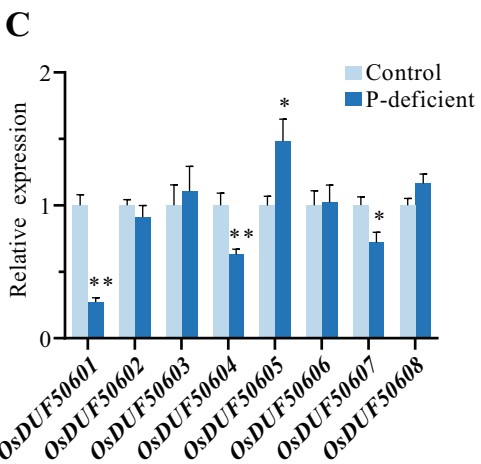

**Figure 10 Relative gene expressions of *OsDUF506s* under drought (A), cold (B) and phosphorus-deficient (C) stresses.** Data represent the mean ± SE of three biological and technical replicates. The significant differences level was analyzed by unpaired t-test ($^*p < 0.05$, $^{**}p < 0.01$).

gene expression profiles of the RiceXPro database (Fig. 8) and Rice Genome Annotation Project database (http://rice.uga.edu/expression.shtml), they were barely expressed in any tissue. Besides, compared to other members, *OsDUF50609* and *OsDUF50610* contained the most nsSNP mutations (Table 2), and excessive nonsynonymous nucleotide mutants are a characteristic of pseudogenes (*Balakirev & Ayala, 2003*). DUF family appears to be abundant in pseudogenes, such as DUF1311, DUF 1124, and DUF 3054 (*Thibaud-Nissen, Ouyang & Buell, 2009*). On the other side, *OsDUF50609* and *OsDUF50610* exhibited a general loss bias in Gramineae species, including *Zea mays*, *Hordeum vulgare*, *Triticum aestivum*, and *Sorghum bicolor* (Fig. 6). Except for the presumed pseudogenes, the other *OsDUF506*s all have colinear gene pairs with the other five monocots (Fig. 6), revealing that they were conserved in the evolution and expansion of the *DUF506* family in plants and they might have originated from the same ancestor. Previous studies suggested that more than half of duplicated genes have diverged in gene expression in both *Arabidopsis*

and *Oryza sativa* (*Blanc & Wolfe, 2004*; *Yim, Lee & Jang, 2009*). Two duplicated gene pairs from subfamily IIIb (*OsDUF50604-OsDUF50608* and *OsDUF50605-OsDUF50607*) revealed different tissue specificity (Fig. 8), and duplicates respond conversely under phosphorus-deficient conditions (Fig. 10), suggesting that they might be neofunctionalized.

MicroRNAs (miRNAs) are fundamental noncoding riboregulators for gene expression. In plants, miRNA silences genes by guiding RNA cleavage or translation inhibition (*Song et al., 2019*). They cooperate closely with target genes and transcription factors to regulate plant growth and resistance; it could be an effective strategy for precisely improving *Oryza sativa* varieties by derepressing specific genes using CRISPR/Cas9 (*Lin et al., 2021*; *Nadarajah & Kumar, 2019*). We predicted the miRNAs targeting *OsDUF506s* and analyzed their expression profiles in tissues and under stresses of drought, cold, and salt (Fig. 7). The osa-miR444b.1, which specifically targeted *OsDUF50606*, was the only one highly expressed in tissues excluding the anthers and under the four stresses. It was functionally unknown. One copy of the segmental duplicates from subfamily II (*OsDUF50606*) expressed extremely low in all tissues, it showed no significant changes in expression level under those abiotic stresses. In contrast, the other copy (*OsDUF50601*) was constitutively expressed and was strongly induced by drought, cold, and phosphorus-deficient stresses (Figs. 8 and 10). Further verification is required to ascertain if the expression difference originates from the translation inhibition by osa-miR444b.1.

The results of expression induced by plant hormones showed that *OsDUF506s* were more sensitive to ABA and JA treatments than IAA and GA$_3$ treatments, which was consistent with the presence of hormone-responsive CREs observed in their promoters (Figs. 3 and 9). Although the promoter region of some *OsDUF506s* contained a few GA$_3$ and IAA CREs, they show no significant and specific response trend under corresponding hormone treatment.

The CREs analysis also showed that *OsDUF50601*, *OsDUF50602*, *OsDUF50607*, and *OsDUF50609* each contained one MBS, which was a MYB transcription factor binding site responsive to drought. However, only *OsDUF50602* was upregulated under drought treatment (Figs. 3 and 10), indicating that their promotors might recruit different MYBs to active *OsDUF50602* expression and inhibit *OsDUF50601* and *OsDUF50607* expressions to regulate drought tolerance. Perhaps this is due to the vital function of hormones under the drought condition. ABA is an important plant hormone regulating water status and stomatal movement. When plants suffer from a drought environment, they synthesize ABA. Increasing ABA could induce plants to close their stomatal and retain water (*Lim et al., 2015*). Seven copies of ABRE(ABA-responsive element)existed in the promotor region of *OsDUF50602*. Its expression significantly increased in both shoot and root under ABA treatment, suggesting that it might also be involved in the ABA-related signaling pathway of drought responses (Fig. 9). Moreover, a previous study revealed that the expression of *OsDUF50602* was higher in *OsDT11* OE lines than in the wild line under drought treatment, indicating that the drought response of *OsDUF50602* might be enhanced in the particular genetic background (*Zhao et al., 2020*). ABRE has been proved to be the most conserved drought-inducible promoter in Arabidopsis, *Oryza sativa*, and

soybean, indicating that the transcriptional regulation of drought-inducible genes like *OsDUF50602* is similar across these species (*Maruyama et al., 2012*). Therefore, we analyzed the expressions of its homologous genes under drought stress in the eplant database (http://bar.utoronto.ca/). In *Triticum aestivum*, TraesCS3A02G420900 gene expression increased 3.34-fold under drought stress. On the other hand, JA and its derivatives, which occur at low levels under normal condition, accumulate to high levels and are transmitted over long distances under abiotic stress (*Wang et al., 2021a*). The promoter of *OsDUF50602* contained six copies of the CGTCA/TGACG-motif (JA-responsive element). With JA treatment, the expression of *OsDUF50602* was rapidly and significantly upregulated in both the shoot and root (Fig. 9). However, further verification is required to determine whether these two hormones collaboratively induce the drought responses of *OsDUF50602*.

The *OsDUF506s* were more sensitive to cold stress than to drought stress. Under cold stress, except for the slight downregulation of *OsDUF50606* and *OsDUF50607*, the other *OsDUF506s* were significantly upregulated, and three genes containing LTR elements (*OsDUF5060601*, *OsDUF5060604*, and *OsDUF50608*) showed the highest level (Fig. 10). *DUF506s* from other species also revealed an active response to cold stress. For instance, *At1g62420*, *At3g25240*, *Bradi2g58590*, and *Bradi2g62310* were strongly induced by cold stress in *Arabidopsis* and *Brachypodium* (*Ying, 2021*). The above results demonstrated that *DUF506* genes play a crucial role in cold response with different mechanisms.

Recent studies showed five *AtDUF506* genes belonging to subfamilies I and II (*At1g62420*, *At3g07350*, *At3g25240*, *At2g20670*, and *At4g32480*) were strongly upregulated by P-limitation (*Ying et al., 2022*; *Ying & Scheible, 2022*). However, although the *OsDUF506s*, which belong to subfamilies I and II, shared a highly similar exon/intron structure and conserved motifs (Fig. 3), only *OsDUF50601* was significantly downregulated by P-limitation, indicating that monocot and dicot species differed in phosphorus responding signal pathway.

Rice (*Oryza sativa*) is an essential staple food for approximately half of the world's population, and stable rice production is crucial for food security, especially in Asia (*Zhang, 2007*). However, extreme weather and climate change, such as drought, flooding, salinity, low temperature, high temperature, and mineral deficiency, seriously affect crop productivity and sustainability. Among the abiotic stress, drought is the most destructive threat. Half of the world's arable land will suffer from drought in the next three decades (*Singhal et al., 2016*). Hence, it is urgent to breed rice varieties with superior drought resistance. Drought resistance is a complex agronomic trait regulated by multiple genes. Exploring drought resistance genes through linkage analysis or GWAS is inefficient and genes with minor effects are hard to clone (*Sun et al., 2022*). With the development of bioinformatics and gene editing technology, the reverse genetics method can be combined to identify and characterize abiotic resistance-related genes such as *OsDUF506s*. Abiotic resistance improvements usually require increased expression of related genes, including *DROUGHT1* (*DROT1*), *ONAC066*, *OsMADS23*, *DROUGHT-INDUCED BRANCHED-CHAIN AMINO ACID AMINOTRANSFERASE* (*OsDIAT*), *CHILLING-TOLERANCE DIVERGENCE 1* (*COLD1*), and *Cyclic Nucleotide-gated Channels 9*

(*OsCNGC9*) (*Sun et al., 2022*; *Yuan et al., 2007*; *Li et al., 2021*; *Shim et al., 2023*; *Ma et al., 2015*; *Wang et al., 2021b*). A donor-DNA-free CRISPR/Cas-based approach to knock-up genes in rice could be helpful for rice breeding practices and simplify regulatory approval of the edited plants (*Lu et al., 2021*).

## CONCLUSIONS

In this study, 10 *OsDUF506* family members in *Oryza sativa* were identified and classified into four subfamilies. We analyzed the phylogenetic relationship, gene structures, conserved motif, CREs, SNP distribution, and targeting miRNA, which filled the gap of *DUF506* family in rice. The results of public expression profiles and RT-qPCR data in tissues and under plant hormones and abiotic stresses demonstrated that *OsDUF506s* were actively involved in ABA and JA response and had different expression patterns under drought and cold, which laid the foundation for further functional analysis of *OsDUF506* family.

### Funding

This work was supported by the Yunnan Province Major Science and Technology Project (No. 202102AE090016, No. 202302AE090001, No. 202302AE090003), and the Yunnan Province Scientific and Technological Talent and Platform Projects (No. 202105AE160009, No. 202305AF150195, Title. Colored Rice Varieties Breeding and Green Technologies Development Project). The funders had no role in study design, data collection and analysis, decision to publish, or preparation of the manuscript.

### Grant Disclosures

The following grant information was disclosed by the authors:
Yunnan Province Major Science and Technology: 202102AE090016, 202302AE090001, 202302AE090003.
Yunnan Province Scientific and Technological Talent and Platform: 202105AE160009, 202305AF150195.

### Competing Interests

Kui Fan, Rui Wang & Jianping Zhang are employed by Yunnan Grain Industry Group Co., Ltd.

### Author Contributions

- Wei Dong conceived and designed the experiments, performed the experiments, analyzed the data, prepared figures and/or tables, and approved the final draft.
- Jian Tu performed the experiments, analyzed the data, prepared figures and/or tables, and approved the final draft.
- Wei Deng performed the experiments, prepared figures and/or tables, and approved the final draft.

- Jianhua Zhang performed the experiments, prepared figures and/or tables, and approved the final draft.
- Yuran Xu performed the experiments, prepared figures and/or tables, and approved the final draft.
- Anyu Gu analyzed the data, prepared figures and/or tables, and approved the final draft.
- Hua An analyzed the data, prepared figures and/or tables, and approved the final draft.
- Kui Fan analyzed the data, authored or reviewed drafts of the article, and approved the final draft.
- Rui Wang performed the experiments, authored or reviewed drafts of the article, and approved the final draft.
- Jianping Zhang analyzed the data, authored or reviewed drafts of the article, and approved the final draft.
- Limei Kui conceived and designed the experiments, authored or reviewed drafts of the article, and approved the final draft.
- Xiaolin Li conceived and designed the experiments, authored or reviewed drafts of the article, and approved the final draft.

## Data Availability

The raw data is available in the Supplemental File.

## Supplemental Information

Supplemental information for this article can be found online at http://dx.doi.org/10.7717/peerj.16168#supplemental-information.

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
