# Peer review of "Genome-wide identification of DUF506 gene family in Oryza sativa and expression profiling under abiotic stresses"

_PeerJ, doi:10.7717/peerj.16168_

## Round 0.1 · original submission · Major Revisions

The authors put considerable effort into this study by identifying and classifying the OsDUF506 gene family in rice under drought, cold, and phosphorus deficiency stress. The manuscript was revised by two reviewers who suggested revision of the manuscript. Please edit the manuscript according to the reviewers' suggestions and address the comments point-by-point. In addition, highlight all associated changes made to the manuscript using track changes.

**Language Note:** The review process has identified that the English language must be improved. PeerJ can provide language editing services - please contact us at copyediting@peerj.com for pricing (be sure to provide your manuscript number and title). Alternatively, you should make your own arrangements to improve the language quality and provide details in your response letter. – PeerJ Staff

·

Basic reporting

The manuscript is well written and interesting but still needs some minor changes

Experimental design

The study demonstrates a good experimental design.

Validity of the findings

No comments

Additional comments

I reviewed the paper titled "Genome-wide identiûcation of DUF506gene family in rice and expression proûling under abiotic stresses". The authors provided a more comprehensive identification and classification of OsDUF506s, expanded the recognition of the functions under abiotic stresses, and served as the basis of molecular breeding in rice.
-Comments and Suggestions for Authors
Abstract
The abstract is well written.
Introduction
-The introduction section is comprehensive and well written.
-Please find more corrections as track changes in the manuscript pdf file.
Materials and methods
- In the expression analysis of OsDUF506 members by qRT-PCR, I suggest the authors to mention the size of amplicon and from either the 5' end or the 3' end the amplicons were amplified
-Please find more corrections as track changes in the manuscript pdf file.
Results
-The results section is well written.
-Line 190, the word "was" could be replaced with "were" in the first sentence, since the subject "130 DUF506s" is plural.
-Line 295, the word "expressing" could be replaced with "expression" to make the sentence more concise.
- Line 296, The word "showed" could be replaced with "showed no" to make the sentence more clear. Or you can correct it as " did not show significant changes".
- Line 315 … such as…
- Please find more corrections as track changes in the manuscript pdf file.

Discussion
-The discussion section is well written.

Conclusion
-The conclusion section is well written.
References
Please unify the style according to the journal instructions
Figure 1: Please add more information such as " Bootstrap values were calculated from 1000 replications and only the values with ?????? % bootstrapping were considered significant, and are indicated on the branch nodes. "
Figure 2: Please rewrite as the following "Green bars,yellow bars, and lines indicate UTRs, exons, and introns, respectively."

Reviewer 2 ·

Basic reporting

Languages required to be polished by a fluent English speaker.

Experimental design

See "Additional comments"

Validity of the findings

See "Additional comments"

Additional comments

The authors of this manuscript aimed to comprehensively identify and characterize the members of Domain of Unknown Function 506/DUF506 family in rice. In silico analysis, such as phylogenetic, evolutionary and synteny analysis were implemented. In addition, transcriptional analysis of OsDUF506s have been conducted by exploring public data inventories and RT-qPCR. Overall, the design and analysis of this study are sound, and the results are exhaustive. However, some results are required to re-examine or re-analyze. Below, I outlined some of my major concerns that need to be addressed before next submission.
INTRODUCTION
Line 68-69: The authors have stated that DUF506 protein have not been functionally characterized. However, Ying et al. (2022) have demonstrated that two distinct Arabidopsis DUF506 protein regulated root hair growth. Thus, the statement needs to be rephrased.
Line 77-79: AtRXR3 is not the homolog of AtRXR1. Also, AtRXR3 is transactivated by RSL4 to regulate root hair growth (Ying and Scheible, 2022).
The full name of CaM and OsDT11 should be provided.
METHODS AND MATERIALS
Line 104: “10 species” need to provide the details.
Line 163-169: the receipt of Yoshida nutrient solution (YNS) should be detailed. The description of culture condition is incomplete, such as light intensity and relative humidity. 20% PEG-6000, w/v or v/v??? what is the P concentration in “phosphate-deficient YNS”?
RESULTS
Major issue - Ying (2021) extensively explored DUF506 family in 17 plant species, some of which overlapped with the species that used in this study, such as rice, maize, soybean and barely. Interestingly, the members of DUF506s in this study is inconsistent with previous study. For instance, Ying (2021) identified 11 DUF506 proteins in rice, including LOC_Os1g65740 and LOC_Os2g48850, which were not present in this study. These discrepancies might be caused by different selection criteria. The authors should explicitly clarify the differences.
Fig 2A is not mentioned in the manuscript.
In Ying et al. (2022), three conserved motif/domain have been identified from Arabidopsis DUF506 proteins. Are these motifs also observed in rice DUF506s? Moreover, where is the PDDEXK signature motif localized?
Line 222-223: does the 2000 bp upstream of DUF506 start from the transcriptional or translational site?
Fig 5: I don’t think that At4g14620 collineates with OsDUF50604 and OsDUF50607, because they were not a duplicated pair.
Fig 7 and Table S10, several different miRNA had identical expression level. Authors need to explain this.
Fig 10, authors examined transcript changes of 8 OsDUF506 genes in shoot under drought, cold and P-deficient stress. Why are the OsDUF50609 and OsDUF50610 missing? It is of interest to know their expression pattern in root. Because OsDUF50605 and OsDUF50607, the duplicated pair, exhibited opposite expression pattern in shoot and root, when exposed to ABA treatment (fig 9).
To assess the extend of abiotic stress, it would be helpful including a marker gene to demonstrate the particular treatment is effective. For instance, using RAB18 gene to validate ABA treatment. I suggest that authors test a few marker genes, especially for P-deficient treatment, to support their conclusion.
DISCUSSION
Line 352: 130 DUF506 genes
Line 357-361: the findings such as the divergent group of DUF506 proteins between monocot and dicot, and relationship of genome size and DUF506 family, have been discovered and discussed in Ying (2021). Authors should structure the discussion based on their own research.
The OsDUF50602 gene impressed me mostly, not only because of its remarkable responses to different abiotic stress and phytohormone treatment (fig 9 and 10), but also because it exhibited similar expression pattern in other species. Authors should focus on exploring the related cis-regulatory elements in its promoter. For instance, does the high abundance of stress related CREs lead to the dynamic expression changes.
Authors should provide some insightful directions for future research based on the current findings. For instance, how to leverage these rice DUF506 gene through genetic engineering to enhance crop tolerance to increasingly severe climate changes.

---

## Round 0.2 · accepted · Accept

Dear Authors,

Your revised manuscript and response letter have been sent to the two reviewers. Both reviewers accepted your manuscript in its current revised status. Accordingly, I accept your revised manuscript and suggest its publishing in PeerJ.

Kind regards
Elsayed Mansour

The Section Editor also noted:

> I didn't see anywhere in the text where "rice" was identified by genus/species, only in the figure legends. This should be changed thoughout the manuscript.

·

Basic reporting

no comment

Experimental design

no comment

Validity of the findings

no comment

Additional comments

The authors have made the changes I suggested in the last review. I recommend the publication of the article in this journal.

Reviewer 2 ·

Basic reporting

The authors have clearly addressed all my concerns. I have no further questions.

Experimental design

See below

Validity of the findings

See below

Additional comments

I suggested adding figure 3 and 4 (in the Rebuttal letter) to the supplemental information to support the main conclusions.

Minor issue:
Line 24: DUF506, not DUF560.